# Transforming to Yoked Neural Networks to Improve ANN Structure

## Abstract

Most existing classical artificial neural networks (ANN) are designed as a tree structure to imitate neural networks. In this paper, we argue that the connectivity of a tree is not sufficient to characterize a neural network. The nodes of the same level of a tree cannot be connected with each other, i.e., these neural unit cannot share information with each other, which is a major drawback of ANN. Although ANN has been significantly improved in recent years to more complex structures, such as the directed acyclic graph (DAG), these methods also have unidirectional and acyclic bias for ANN. In this paper, we propose a method to build a bidirectional complete graph for the nodes in the same level of an ANN, which yokes the nodes of the same level to formulate a neural module. We call our model as YNN in short. YNN promotes the information transfer significantly which obviously helps in improving the performance of the method. Our YNN can imitate neural networks much better compared with the traditional ANN. In this paper, we analyze the existing structural bias of ANN and propose a model YNN to efficiently eliminate such structural bias. In our model, nodes also carry out aggregation and transformation of features, and edges determine the flow of information. We further impose auxiliary sparsity constraint to the distribution of connectedness, which promotes the learned structure to focus on critical connections. Finally, based on the optimized structure, we also design small neural module structure based on the minimum cut technique to reduce the computational burden of the YNN model. This learning process is compatible with the existing networks and different tasks. The obtained quantitative experimental results reflect that the learned connectivity is superior to the traditional NN structure.

## 1 Introduction

Deep learning successfully transits the feature engineering from manual to automatic design and enables optimization of the mapping function from sample to feature. Consequently, the search for effective neural networks has gradually become an important and practical direction. However, designing the architecture remains a challenging task. Certain research studies explore the impact of depth [1,2,3] and the type of convolution [4,5] on performance. Moreover, some researchers have attempted to simplify the architecture design. VGGNet [6] was directly stacked by a series of convolution layers with plain topology. To better adapt the optimization process of gradient descent process, GoogleNet [7] introduced parallel modules, while Highway networks [8] employed gating units to regulate information flow, resulting in elastic topologies. Driven by the significance of depth, the residual block consisted of residual mapping and shortcut was raised in ResNet [9]. Topological changes in neural networks successfully scaled up neural networks to hundreds of layers. The proposed residual connectivity was widely approved and was subsequently applied in other works such as MobileNet [10,11] and ShuffleNet [12]. Divergent from the relative sparse topologies,

DenseNet [13] wired densely among blocks to fully leverage feature reuse. Recent advances in computer vision [25,26] also explored neural architecture search (NAS) methods [14,15,16] to search convolutional blocks. In recent years, Yuan proposed a topological perspective using directed acyclic graph (DAG) [29] to represent neural networks, enhancing the topological capabilities of artificial neural networks (ANNs). However, these approaches suffer from the bias of unidirectional and acyclic structures, limiting the signal's capability for free transmission in the network.

The existing efforts in neural network connectivity have primarily focused on the tree structures where neural units at the same level cannot exchange information with each other, resulting in significant drawbacks for ANNs. This limitation arises due to the absence of a neural module concept. In this paper, we argue that the current connectivity approaches fail to adequately capture the essence of neural networks. Since the nodes at the same level of a tree cannot establish connections with each other, it hampers the transfer of information between these neural units, leading to substantial defects for ANNs. We argue that the nodes in the same level should form a neural module and establish interconnections. As a result, we introduce a method to build up a bidirectional complete graph for nodes at the same level of an ANN. By linking the nodes in a YOKE fashion, we create neural modules. Furthermore, when we consider all the nodes at the same level, we would have a chance to construct a bidirectional complete graph in ANNs and yields remarkable improvements. We refer to our model as Yoked Neural Network, YNN for brevity. It is important to note that if all the edge weights in the bidirectional complete graph become vestigial and approach to zero, our YNN would reduce to a traditional tree structure.

In this paper, we analyze the structural bias of existing ANN structures. To more accurately mimic neural networks, our method efficiently eliminates structural bias. In our model, nodes not only aggregate and transform features but also determine the information flow. We achieve this by assigning learnable parameters to the edges, which reflect the magnitude of connections. This allows the learning process to resemble traditional learning methods, enhancing the overall performance of our model in imitating neural networks. As the nodes are relied on the values of other nodes, it is a challenging task designing a bidirectional complete graph for nodes at the same level. We address this challenge by introducing a synchronization method specifically tailored for learning the nodes at the same level. This synchronization method is crucial for ensuring the effective coordination and learning of these interconnected nodes.

Finally, to optimize the structure of YNN, we further attach an auxiliary sparsity constraint that influences the distribution of connectedness. This constraint promotes the learned structure to prioritize critical connections, enhancing the overall efficiency of the learning process.

The learning process is compatible with existing networks and exhibits adaptability to larger search spaces and diverse tasks, effectively eliminating the structural bias. We evaluate the effectiveness of our optimization method by conducting experiments on classical networks, demonstrating its competitiveness compared to existing networks. Additionally, to showcase the benefits of connectivity learning, we evaluate our method across various tasks and datasets. The quantitative results from these experiments indicate the superiority of the learned connectivity in terms of performance and effectiveness.

Considering that the synchronization algorithm for nodes at the same level may be computationally intense, we also propose a method to design small neural modules to simplify our model. This approach significantly reduces the computational burden of our model while maintaining its effectiveness.

To sum up, our contributions in this paper are as follows:

1. We provide an analysis of the structural bias present in existing ANN networks.

2. We propose the YNN model which involves YOKING the nodes at the same level together to simulate real neural networks.

3. We develop a synchronization method to effectively learn and coordinate the nodes at the same level, introducing the concept of neural modules.

4. We design a regularization-based optimization method to optimize the structure of the YNN model.

5. We propose the design of small neural modules to significantly reduce the computational complexity of our model, improving its efficiently.

## 2  Related Works

We firstly review some related works on the design of neural network structures and relevant optimization methods. The design of neural network has been studied widely. From shallow to deep, the shortcut connection plays an important role. Before ResNet, an early practice [17] also added linear layers connected from input to output to train multi-layer perceptrons. [7] was composed of a shortcut branch and a few deeper branches. The existence of shortcut eases the vanishing or exploding gradients [8,9]. Recently, Yuan [29] explained from a topological perspective that shortcuts offer dense connections and benefit optimization. Many networks with dense connections exist On macro-structures also. In DenseNet [13], all preceding layers are connected. HRNet [18] was benefited from dense high-to-low connections for fine representations. Densely connected networks promote the specific task of localization [19]. Differently, our YNN optimizes the desired network from a bidirectional complete graph in a differentiable way.

For the learning process, our method is consistent with DARTS [22], which is differentiable. Different from sample-based optimization methods [29], the connectivity is learned simultaneously through the weights of the network using our modified version of the gradient descent. A joint training can shift the transferring step from one one task to another, and obtain task-related YNN. This type was explored in [20,21,22,23,24] also, where weight-sharing is utilized across models at the cost of training. At the same time, for our YNN model, we also propose a synchronization method to get the node values in the same neural module.

In order to optimize the learned structure, a sparsity constraint can be observed in other applications, e.g., path selection for a multi-branch network [27], pruning unimportant channels for fast inference [28], etc. In a recent work, Yuan used L1 regularization to optimize a topological structure. In this paper, we also use L1 as well as L2 regularization to search a better structure.

Secondly, many deep learning works deal with the geometric data in these years[40]. They make neural network better cope with structure. Graph neural networks (GNNs) are connectivity-driven models, which have been addressing the need of geometric deep learning[30,31]. In fact, a GNN adapts its structure to that of an input graph, and captures complex dependencies of an underlying system through an iterative process of aggregation of information. This allows to predict the properties of specific nodes, connections, or of the entire graph as a whole, and also to generalize to unseen graphs. Due to these powerful features, GNNs have been utilized in many relevant applications to accomplish their tasks, such as recommender systems [33], natural language processing [34], traffic speed prediction [35], critical data classification [36], computer vision [25,26,37], particle physics [38], resource allocation in computer networks [39], and so on.

## 3  Methodology

### 3.1  Why YNN is Introduced?

NN stands for a type of information flow. The traditional structure of ANN is a tree, which is a natural way to describe this type of information flow. Then, we can represent the architecture as $G = (N, E)$, where $N$ is the set of nodes and $E$ denotes the set of edges. In this tree, each edge $e_{ij} \in E$ performs a transformation operation parameterized by $w_{ij}$, where $ij$ stands for the topological ordering from the node $n_i$ to node $n_j$ with $n_i, n_j \in N$. In fact, the importance of the connection is determined by the weight of $e_{ij}$. The tree structure as a natural way to represent such formation flow is most frequently used in ANN.

A tree is a hierarchical nested structure where a node can be influenced only by its precursor node, thereby causing transformation of information between them. In a tree structure, the root node has no precursor node, while each other node has one and only one precursor node. The leaf node has no subsequent nodes. The number of subsequent nodes of each other node can be one or multiple. In addition, the tree structure in mathematical statistics can represent some hierarchical relationships. A tree structure has many applications. It can also indicate subordinating relationships.

In recent years, some researchers attempted to generalize this structure. In those works, except the root node, all other nodes are made to have multiple precursor nodes, i.e., the hierarchical information flow is made to form a directed acyclic graph (DAG).

However, a tree or a DAG is a hierarchical nested structure where a node can be influenced only by its precursor node, which makes the transformation of information quite inadequate. Moreover, we find that this structure is far more inferior in its strength compared with those of real neural networks, which connect far more complex structures than a tree or DAG structure as shown in Fig 1. In fact, a tree or a DAG structure is used just because its good mathematical properties which can apply backward propagation conveniently.

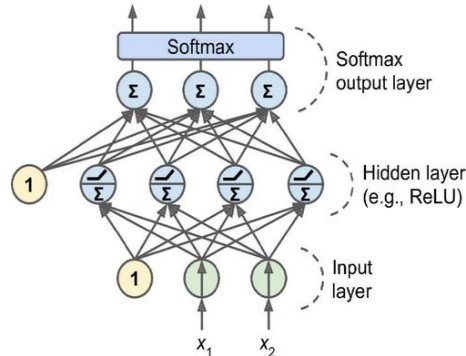

Figure 1: Artificial Neural Network

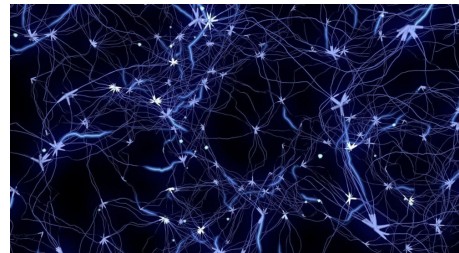

Figure 2: Real Neural Network

In this paper, we represent the neural network as a bidirectional complete graph for the nodes of the same level to make the description of NN is much better compared with the traditional ANN. Further, the connections between nodes are represented as directed edges, which determine the flow of information between the connected nodes. We consider that any two nodes $n_i$ and $n_j$ of the same level construct an information clique if there exists a path between them. Compared with the traditional tree structure, we yoke the nodes of the same level to form a bidirectional complete graph. We call this structure as YNN, which will be introduced in the next section.

## 3.2 Structure of YNN

Inspired by the neural network of human beings as shown in the Fig 2. In order to enhance the ability of NN to express information, we design cliques for the nodes of each level of a neural network.

**Definition 1** *A clique is a bidirectional complete graph which considers that for any two nodes $n_i$ and $n_j$, an edge exists from $n_i$ to $n_j$.*

According to this definition, the model in our framework is considered as a bidirectional complete graph for the nodes of the same level. These nodes construct a clique, where every node is not only influenced by its precursor nodes but also by all other nodes of its level. The cliques are represented as information modules which greatly enhance the characterization of NN.

According to the definition of clique, a neural network can also be represented as a list of cliques. Further, we can also introduce a concept of neural module.

**Definition 2** *A neural module is a collection of nodes that interact with each other.*

According to the definition, a neural module can be part of clique. In fact, if all the weights in a clique becomes zero, then the YNN model is reduced to the traditional tree structure.

In each clique of our model, the nodes are first calculated by using their precursor nodes, which only distribute features. The last one is the output level, which only generates final output of the graph. Secondly, each node is also indicated by the nodes of the same level and their values are influenced by each other.

During the traditional forward computation, each node aggregates inputs from connected preorder nodes. We divide such nodes into two parts. The first part contains the precursor nodes of the last level, and the second part contains the nodes of the corresponding clique of the same level. Then, features are transformed to get an output tensor, which is sent to the nodes in the next level through the output edges. Its specific calculation method will be introduced in the next section.

178 In summary, according to the above definitions, each YNN is constructed as follow. Its order of
179 outputs is represented as $G = \{N, E\}$. For the nodes in the same level, bidirectional complete graphs
180 are built as clique $C$. Each node $n$ in $C$ is first calculated by using the precursor nodes without the
181 nodes in the clique, which is called as the meta value $\hat{n}$ of the node. Then, we calculate its real value
182 $n$ by using the nodes of the clique.

183 According to the meta value and the real value as introduced before, the structure of YNN is shown
184 in the Fig 3.

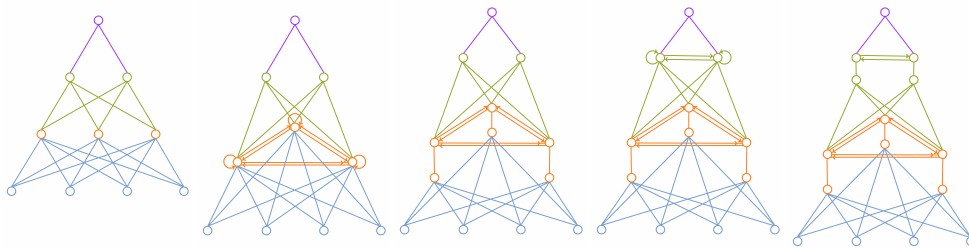

Figure 3: The first picture shows the tree structure of traditional ANN. The second picture shows our
YNN model that yokes together the nodes of the first level. For the clique of the first level, the node
spin part is based on its meta value, which also represents the connection with the pre nodes. As a
result, we can decompose the spin node as shown in the third picture, which is to represent the meta
value. The fourth and fifth pictures show the second level of our YNN model, which are the same as
the second and third pictures, respectively.

185 In the next section, we will explain how to calculate the values of the nodes by using the precursor
186 node as well as the nodes in the clique.

## 3.3 Forward Process

188 Let we have $n$ elements:
$$X = \{x_1, x_2, ..., x_n\} \tag{1}$$
189 as the input data to feed for the first level of ANN. Then, the meta value $\widehat{N}^1$ of the first level can be
190 calculated as:
$$\widehat{N}^1 = X * W^{01}, \tag{2}$$
191 where $W_{01}$ is the fully connected weight of the edges between level 1 and input nodes. Then,
192 similarity in nature, for meta value, the full connection between the levels makes the information to
193 flow as:
$$\widehat{N}^i = f(N^{i-1}) * W^{(i-1)i}, \tag{3}$$
194 where $N^{i-1} = \{1, n_1^{i-1}, n_2^{i-1}, ...\}$, $n_j^{i-1}$ is the real value of the $j$th node in the $(i-1)$th level,
195 number 1 indicates for the bias of the value between the $(i-1)$th and $i$th levels as well as the
196 activation function $f$.

197 Then, by introducing weight $W^i$ in the $i$th level and considering the bidirectional complete graph of
198 that level as a clique, we propose a method to calculate the real value $N^i$ based on the meta value $\widehat{N}^i$
199 as introduced in the previous section. Suppose, there are $m$ nodes in the clique and they rely on the
200 values of other nodes. Hence, we need a synchronization method to solve the problem. Here, we take
201 the problem as a system of multivariate equations as well as an activation function $f$. Then, for the
202 real value of $n_j^i$ in $N^i$ based on the meta value $\widehat{n}_j^i$ in $\widehat{N}^i$, the equations can be summarized as follow:

$$\begin{cases} w_{01}^i + \sum_{j \neq 1} f(n_j^i) * w_{j1}^i + f(\widehat{n}_1^i) * w_{11}^i = n_1^i \\ w_{02}^i + \sum_{j \neq 2} f(n_j^i) * w_{j2}^i + f(\widehat{n}_2^i) * w_{22}^i = n_2^i \\ ... \\ w_{0m}^i + \sum_{j \neq m} f(n_j^i) * w_{jm}^i + f(\widehat{n}_m^i) * w_{mm}^i = n_m^i \end{cases}$$

In the above equations, $w_{01}^i, w_{02}^i, ..., w_{0m}^i$ are the bias of the real values of the nodes in the $i$th level. Note that, for the mata value, the bias is a value between the levels; while for a real value, the bias is a value in the individual level only.

Existing numerical methods would be able to solve the above equations efficiently. In the real applications, the efficiency can also be well optimized. In fact, for too large equations, we also propose a method to reduce the calculation scale efficiently. This method is introduced in the following section.

### 3.4 Backward Process

In this section, we introduce the backward process of our model. Firstly, let the gradient of the output be the gradient of the meta value of the last level. We calculate the node gradient for the $i$th level as:

$$d(N^i) = d(\widehat{N}^{i+1}) * W^{i(i+1)T} * f^{-1}(N^i) . \tag{4}$$

The meta value of $\widehat{N}^i$ is calculated by using the real value of $N^{i-1}$ according to the system of equations.

Then, to get the value of $d(\widehat{N}^{i-1})$, we need to consider the nodes as the variables in the system of equations. For convenient, we introduce operator $C^i$ to represent the derivatives for the $i$th level, which can be expressed as:

$$C^i = W^i - diag(W^i) + eye(W^i) , \tag{5}$$

where $W^i$ is the adjacency matrix of the clique in the $i$th level, $diag(W^i)$ is the diagonal matrix of $W^i$, $eye(W^i)$ is the identity matrix whose size is the same as that of $w^i$, and operator $C^i$ represents the transfer of other nodes for each node in the clique according to the system of equations. In the clique, the identity matrix is for the node itself.

According to the system of equations, the meta value of a node is connected to its real value through the diagonal matrix of $W^i$. Note that each node is calculated by using the activation function $f$. As a result, after the transfer through the bidirectional complete graph, the gradient of the meta value of the nodes becomes:

$$d(\widehat{N}^i) = d(N^i) * C^{iT} * f^{-1}(N^i) * diag(W^i) * f^{-1}(\widehat{N}^i) . \tag{6}$$

Now, we have got the gradient of the meta value as well as that of the real value of each node. Finally, the gradient weight of the fully connected level $W^{i(i+1)}$ between the $i$th and $(i + 1)$th level can be expressed as:

$$d(W^{i(i+1)})^T = d(\widehat{N}^{i+1})^T * f(N^i) . \tag{7}$$

Now, we need to calculate the gradient of $W^i$ for the clique in the $i$th level. According to the system of equations, we need to consider the weights of all the connected nodes. For any $j$th node in the clique, its connected weight is the $j$th column of the matrix. Similarly, for convenient, we introduce the following operator:

$$D_j^i = (n_1^i, ..., \widehat{n}_j^i, ..., n_m^i) , \tag{8}$$

which can be found in the system of equations. Then, by the gradient of real value of the $j$th node $n_j^i$ in $N^i$, the following becomes the corresponding gradient of the clique:

$$d(W^i(:, j))^T = d(n_j^i) * f(D_j^{i'}) . \tag{9}$$

### 3.5 YNN Structure Optimization

Consider that for the nodes in the same level, we construct a clique as stated before. Here, we consider a clique just as a universal set for all the possible connections. In our work, we can optimize the YNN structure to let our model to focus on important connections only. The optimization process can be L1 or L2 regularization as usual, which can be parameterized $L_1$ and $L_2$, respectively.

For the $j$th node in the $i$th level, the process can be formulated as follow:

$$opt\_n_j^i = n_j^i + L_1 * \sum_k abs(w^i(k, j)) + L_2 * \sum_k (w^i(k, j))^2 \tag{10}$$

According to the L1 and L2 regularization, the L1 parameter can make our YNN to focus on important connections in the clique, and the L2 regularization makes the weight in the clique to be low to make our model to have better generation.

### 3.6 Structure of Neural Module

According to the forward process of YNN as stated earlier, it solves a system of equations. A large number of nodes in the same level would bring too much computational burden to solve a large system of equations. In Fact, we can optimize the graph of any level by L1 and L2 regularization, and then turn to a minimum cut technology, e.g., the NE algorithm, to reduce the computation significantly. For each cut subgraph, we design a neural module structure according to definition 2 to simplify the system of equations as shown in Fig. 4. Since the nodes are influenced only by the nodes in the subgraph, the system of equations can be reduced to the number of the nodes in the cut subgraph, which is formulated as a neural module as definition 2 in this paper.

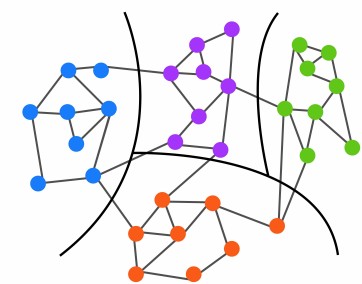

Figure 4: If the clique is too large, we would have too much computational burden to solve the system of equations. Then, we can first optimize the structure and learn the importance of the connection, followed by the application of the minimum cut method to formulate the structure of the neural module. In this way, the calculation for the system of equations can be limited to each subgraph.

In summary, the structure of the neural module can be constructed as follows:

1. Construct the clique for the nodes in the same level;

2. Optimize the clique by using the L1 and L2 regularization;

3. Cut the optimized graph using the NE algorithm;

4. Construct system of equations by taking each cut subgraph as a neural module.

As explained before, in this way the system of equations can be reduced to Ns-ary equations, where $Ns$ is the number of nodes in each neural module. Of course, if the calculation of our model can be accept for our model, take the clique itself as Neural Module is most accurate, since clique considers all connection in the level.

## 4 Experiments

### 4.1 Optimization of Classical ANN

In this section, we will show the experiments with our method. Here, we compare our method with the traditional NN method, stacked auto encoder(SAE), as well as the generalized traditional NN which is a topological perspective to take NN as a DAG graph proposed in recent years.

We show our results for three real data sets. The first dataset contains the codon usage frequencies in the genomic coding DNA of a large sample of diverse organisms obtained from different taxa tabulated in the CUTG database. Here, we further manually curated and harmonized the existing entries by re-classifying the bacteria (bct) class of CUTG into archaea (arc), plasmids (plm), and bacteria

Table 1: Codon Dataset

| Models | Codon Data | | | | | |
|--------|---------|---------|---------|---------|---------|---------|
| | 35 Nodes | 38 Nodes | 40 Nodes | 45 Nodes | 48 Nodes | 50 Nodes |
| NN | 0.248±0.0054 | 0.3098±0.0485 | 0.2815±0.0037 | 0.2664±0.0004 | 0.2837±0.0168 | 0.3955±0.0011 |
| SAE | 0.3446±0.0152 | 0.3282±0.0097 | 0.3588±0.0184 | 0.3294±0.0289 | 0.3055±0.215 | 0.3505±0.0226 |
| DAG | 0.2719±0.0223 | 0.2789±0.0402 | 0.2413±0.0019 | 0.2656±0.0066 | 0.2265±0.0285 | 0.2496±0.0078 |
| YNN | 0.2167±0.0054 | 0.2496±0.0227 | 0.1835±0.0027 | 0.1941±0.0093 | **0.1870±0.0047** | 0.2034±0.0219 |
| YNN&L1 | **0.1999±0.0066** | **0.2117±0.0043** | **0.1706±0.0039** | **0.1846±0.0062** | 0.2132±0.0019 | **0.1839±0.0141** |
| YNN&L2 | 0.2007±0.0137 | 0.212±0.0187 | 0.1816±0.0046 | 0.2085±0.009 | 0.1831±0.0164 | 0.2003±0.0305 |

Table 2: Optical Recognition of Handwritten Digits

| Models | Crowdsourced Data | | | | | |
|--------|---------|---------|---------|---------|---------|---------|
| | 35 Nodes | 38 Nodes | 40 Nodes | 45 Nodes | 48 Nodes | 50 Nodes |
| NN | 0.2565±0.069 | 0.345±0.0011 | 0.2181±0.445 | 0.1536±0.0323 | 0.3159±0.0464 | 0.259±0.0937 |
| SAE | 0.2871±0.04 | 0.2952±0.0209 | 0.3603±0.0086 | 0.4186±0.0419 | 0.3656±0.0228 | 0.3375±0.0376 |
| DAG | 0.2446±0.0409 | 0.2095±0.0014 | 0.2721±0.534 | 0.3475±0.0208 | 0.1981±0.0145 | 0.2585±0.0654 |
| YNN | **0.1433±0.0159** | **0.1274±0.015** | 0.1725±0.0451 | 0.1552±0.0077 | 0.1791±0.0005 | 0.256±0.0001 |
| YNN&L1 | 0.1633±0.0153 | 0.1522$pm$0.0031 | 0.18±0.0247 | 0.1594$pm$0.0225 | **0.143±0.0005** | **0.1494±0.032** |
| YNN&L2 | 0.1586±0.015 | 0.1867±0.186 | **0.1614±0.0189** | **0.1483±0.142** | 0.2028±0.0147 | 0.1881±0.0001 |

proper (keeping with the original label 'bct'). The second dataset contains optically recognized handwritten digits made available by NIST using preprocessing programs to extract normalized bitmaps of handwritten digits from a preprinted form. Out of a total of 43 people. The third dataset is Connect-4 that contains all the legal 8-ply positions used in the game of connect-4, in which neither player has won yet, and the next move is not forced. The outcome class is the theoretical value of the first player in the game.

Here, we compared our method with other methods in terms of a variety of nodes. In this way, we can examine the effectiveness of our model at different levels of complexity of the traditional structure. These nodes are constructed by the NN, SAE, and DAG models. We compared these models in terms of the percentage error. The obtained results are organized in the following Tables, where we can see that our YNN model achieves much better results in most of the cases.

In fact, for all the data sets and a variety of nodes in the same level, our YNN model could tend to get better results after the nodes are yoked together. The effect of our YNN could be improved by optimizing the structure as explained before. All of the first four lines of the Tables are for the results that do not be optimized by the L1 or L2 regularization. We can see that our YNN structure is more efficient even without regularization, compared with the traditional structure.

## 4.2 Optimization of Structure

In this section, we optimize the structure of our model. Since every structure is a subgraph of a fully connected graph, the initial clique can be a search space for our model. Our model is optimized by using the L1 and L2 regularization, which are effective tools for optimizing structures. The obtained results show that such optimizations can yield better effect.

Here, we study the structure of the model for different L1 and L2 parameters, as shown in Fig. 5. In the figure, the green line represents the results of YNN without optimization, while the blue and red lines are the results for a variety of L1 and L2 parameters, respectively. We can see that such optimization is effective for our YNN in most cases.

Table 3: Connect-4 Dataset

| Models | connect-4 Data | | | | | |
|--------|---------|---------|---------|---------|---------|---------|
| | 35 Nodes | 38 Nodes | 40 Nodes | 45 Nodes | 48 Nodes | 50 Nodes |
| NN | 0.2789±0.0075 | 0.2726±0.0099 | 0.285±0.012 | 0.2875±0.0134 | 0.2923±0.0145 | 0.3073±0.0259 |
| SAE | 0.3912±0.0416 | 0.3325±0.0104 | 0.331±0.0044 | 0.3346±0.0096 | 0.3175±0.0082 | 0.3366±0.0099 |
| DAG | 3519±0.05 | 0.2762±0.0038 | 0.2828±0.0053 | 0.2989±0.0081 | 0.3032±0.0009 | 0.3134±0.0382 |
| YNN | **0.2751±0.0174** | 0.265±0.0182 | **0.2489±0.0004** | **0.2582±0.0045** | 0.2569±0.0065 | **0.2475±0.0068** |
| YNN&L1 | 0.2758±0.026 | **0.2544±0.0046** | 0.2513±0.0017 | 0.2635±0.0029 | 0.2574±0.006 | 0.2625±0.0093 |
| YNN&L2 | 0.2826±0.0366 | 0.2577±0.0035 | 0.2495±0.002 | 0.262±0.0081 | **0.2549±0.0067** | 0.2485±0.0122 |

We also show the pixel map of the matrix for the clique. In the figure, the black-and-white graph represents the matrix of the fully connected graph for the nodes in the same level. The more black of the pixel means a lower weight for the corresponding edge.

According with the decline of the error, we can always seek a better structure compared with the bidirectional complete graph used in our YNN. Besides the L1 regularization, the L2 regularization is also an effective tool to optimize the structure of our model. A larger L2 regularization lowers the weights of all the edges, thus yields more black pixels. However, from the decline of error, we can find that the L2 regularization is also effective to optimize our YNN structure.

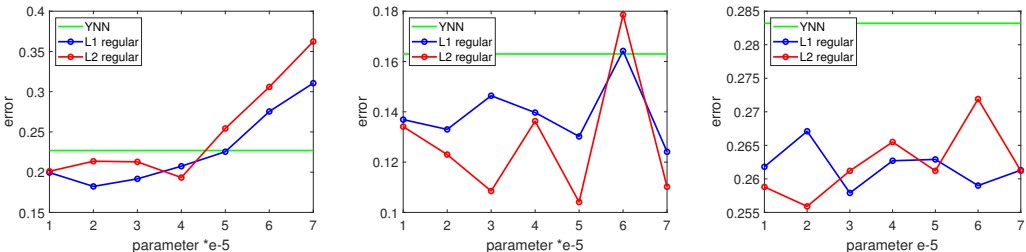

Figure 5: Regularization of results based on L1 and L2 for Codon dataset, optically recognized handwritten digits and connect-4 dataset.

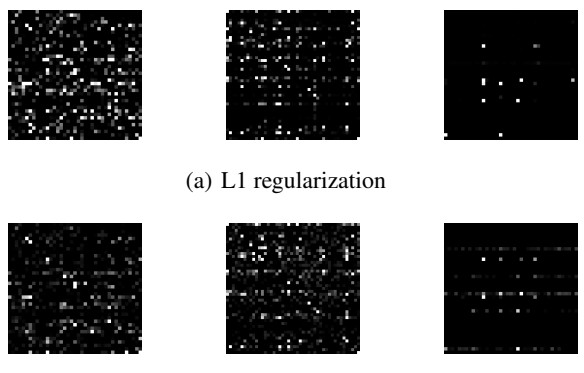

(a) L1 regularization

(b) L2 regularization

Figure 6: Best pixel map of the clique based on L1 and L2 regularization for codon dataset, optically recognized handwritten digits and connect-4 dataset.

## 5 Conclusion

In this paper, we propose a YNN structure to build a bidirectional complete graph for the nodes in the same level of ANN, so as to improve the effect of ANN by promoting the significant transfer of information. In our work, we analyse the structure bias. Our method eliminates structure bias efficiently. By assigning learnable parameters to the edges, which reflect the magnitude of connections, the learning process can be performed in a differentiable manner. For our model, we propose a synchronization method to simultaneously calculate the values of the nodes in the same level. We further impose an auxiliary sparsity constraint to the distribution of connectedness by L1 and L2 regularization, which promotes the learned structure to focus on critical connections. We also propose a small neural module structure that would efficiently reduce the computational burden of our model. The obtained quantitative experimental results demonstrate that the learned YNN structure is superior to the traditional structures.

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
