# OpenReview forum: "Transforming to Yoked Neural Networks to Improve ANN Structure"
_NeurIPS.cc/2023/Conference — Submitted to NeurIPS 2023_

### Official Review · Reviewer_tc8h · 2023-06-27

**Soundness:** 3 good
**Presentation:** 3 good
**Contribution:** 3 good
**Rating:** 5
**Confidence:** 4

**Summary:**

The paper introduces a new method called YNN that transforms traditional ANN structures into yoked neural networks, promoting information transfer and improving performance. The authors analyze the existing structural bias of ANN and propose a model YNN to efficiently eliminate such structural bias. In their model, nodes carry out aggregation and transformation of features, and edges determine the flow of information. They further impose auxiliary sparsity constraints to the distribution of connectedness, which promotes the learned structure to focus on critical connections. Finally, based on the optimized structure, they also design a small neural module structure based on the minimum cut technique to reduce the computational burden of the YNN model. The learning process is compatible with the existing networks and different tasks. The obtained quantitative experimental results reflect that the learned connectivity is superior to the traditional NN structure.

**Strengths:**

1. YNN promotes information transfer significantly which helps in improving the performance of the method.
2. The authors propose a model that efficiently eliminates structural bias in ANN.
3. The authors design a small neural module structure based on the minimum cut technique to reduce the computational burden of the YNN model.

**Weaknesses:**

1. There is a lack of ablation study, e.g., comparing the model performance using different clique size/number of cuts
2. The equations presented in Section 3.3 are excessively complex and challenging to comprehend. The authors have employed "W" and "w" for too many different variables in their notations, leading to confusion.

**Questions:**

The authors seem to be putting forth a novel technique for calibrating hidden states within each layer, which is delineated through the solution of a system of equations, as presented at the bottom of page 5. One might question how this proposed approach contrasts with conventional techniques such as Graph Neural Networks (GNN) or self-attention mechanisms.

Drawing a parallel, it appears that the essence of this newly proposed method bears resemblances to the fundamental principles of GNNs and self-attention methods, notably the propagation of information from one node to its neighboring nodes. Therefore, a comprehensive analysis of the contrasts and similarities of these approaches would be beneficial for a deeper understanding of this proposition.

---

> ### Author Rebuttal · Authors · 2023-08-09
>
> Thank you very much for your precious comments.
>
> 1>Please refer to the pdf file of Author Rebuttal by Authors. I have organized the contribution of the work seriously. I hope it can answer some of your questions. All information is in the picture. If the image is small, please enlarge it.
>
> 2>GNN or self-attention mainly focuses on better organizing the input data to get benefits, e.g. node and edge embedding for GNN and QKV vectors for self-attention. Essentially, by better organizing the input data, they can better handle graph data or sequence data, as self-attention can mine the key information of the sequence. On the other hand, YNN is a generation of the structure of NN from tree to cyclic graph and its benefits can be referred to in the pdf.
>
> 3>The most important forward and backward processes have been carefully organized in the rebuttal pdf of Author Rebuttal by Authors.
>
> 4>Thanks very much.

---

> > ### Author Response · Authors · 2023-08-17
> > **an important message**
> >
> > Dear NeurIPS reviewer,
> >
> > I am writing to draw your utmost attention to our piece of work.
> >
> > At the heart of our innovation lies a critical reimagining of traditional NNs. Currently, NNs operate on asynchronous tensor flow, often organized hierarchically in a tree-like structure. However, this approach inadvertently hampers the nodes within each level from effective communication, relegating them to mere information carriers devoid of meaningful interaction. This inherent limitation substantially diminishes the potential of NNs, impeding their full capabilities.
> >
> > Our work transcends these constraints by introducing a paradigm shift. We present a method that enables synchronous communication among nodes within the same level, a fundamental departure from the status quo. This transformative adjustment yields a remarkable enhancement in information transformation, thereby significantly boosting the overall capacity of NN structures. By fostering a collaborative environment among nodes, our approach leverages their collective power to unlock unprecedented capabilities.
> >
> > Particularly, what sets our research apart is its inspiration drawn from the intricate dynamics of biological neural systems. Unlike the traditional stacked unit approach, where neural elements operate in isolation, our approach mirrors the cooperative nature of biological neural modules. In these systems, multiple neural units collaboratively execute precise functional implementations, resulting in exquisite performance. Our innovation is poised to bridge the gap between artificial and biological neural networks, thus propelling NN structures closer to the remarkable efficiency of their natural counterparts.
> >
> > For a succinct overview of the in-depth details, I encourage you to review the attached one-page PDF in my rebuttal attachment. This document encapsulates the essence of our groundbreaking contribution and underscores the urgency of its consideration. Your attention and support at this juncture are invaluable, and I extend my heartfelt gratitude for your consideration.
> >
> > Warm regards,
> >
> > Authors

---

### Official Review · Reviewer_ZkFh · 2023-07-06

**Soundness:** 2 fair
**Presentation:** 2 fair
**Contribution:** 1 poor
**Rating:** 3
**Confidence:** 4

**Summary:**

This paper proposed a module called YNN that could exchange the information of the neurons within the same layer. The proposed module can be combined with MLP. The experiments on several small-scale datasets show that their method achieves good performance compared to previous networks.

**Strengths:**

- The motivation of this paper is valid and interesting.


**Weaknesses:**

- I don't think a fundamental difference between the proposed YNN and graph neural networks. This method can be a special case by assigning a fully connect adjacent matrix to a GNN.
- The evaluation is only conducted on small-scale datasets.

**Questions:**

- How deep can this YNN be? Typically this kind of network will suffer the information diminishing problem since the feature will be over-smoothed with depth increasing.
- Can this network generalize to large-scale datasets such as imagenet? If so, could the author show some experimental results on it.
- What's the fundamental difference between YNN and GNN?

**Limitations:**

Please refer to the weakness section.

---

> ### Author Rebuttal · Authors · 2023-08-09
>
> Thank you very much for your precious comments.
>
> 1>Please refer to the pdf file of Author Rebuttal by Authors. I have organized the contribution of the work seriously. I hope it can answer some of your questions. All information is in the picture. If the image is small, please enlarge it.
>
> 2>If the model is too deep, as we use the sigmoid activation function, it will suffer from the information diminishing problem. However, typically, we can turn to the relu activation function to alleviate the problem.
>
> 3> According to section 3.6, if we cut to small sub-graphs. Our model model can apply on large data sets such as imagenet.
>
> 4>GNN mainly deals with graph data as input. GNN focuses on better organizing the input data to get benefits, e.g. node and edge embedding for GNN, Essentially, by better organizing the input data, they can better handle graph data. On the other hand, YNN is a generation of the structure of NN from tree to cyclic graph and its benefits can be referred to as the pdf of Author Rebuttal by Authors.
>
> 5>Thanks very much.

---

> > ### Author Response · Authors · 2023-08-17
> > **an important message**
> >
> > Dear NeurIPS reviewer,
> >
> > I am writing to draw your utmost attention to our piece of work.
> >
> > At the heart of our innovation lies a critical reimagining of traditional NNs. Currently, NNs operate on asynchronous tensor flow, often organized hierarchically in a tree-like structure. However, this approach inadvertently hampers the nodes within each level from effective communication, relegating them to mere information carriers devoid of meaningful interaction. This inherent limitation substantially diminishes the potential of NNs, impeding their full capabilities.
> >
> > Our work transcends these constraints by introducing a paradigm shift. We present a method that enables synchronous communication among nodes within the same level, a fundamental departure from the status quo. This transformative adjustment yields a remarkable enhancement in information transformation, thereby significantly boosting the overall capacity of NN structures. By fostering a collaborative environment among nodes, our approach leverages their collective power to unlock unprecedented capabilities.
> >
> > Particularly, what sets our research apart is its inspiration drawn from the intricate dynamics of biological neural systems. Unlike the traditional stacked unit approach, where neural elements operate in isolation, our approach mirrors the cooperative nature of biological neural modules. In these systems, multiple neural units collaboratively execute precise functional implementations, resulting in exquisite performance. Our innovation is poised to bridge the gap between artificial and biological neural networks, thus propelling NN structures closer to the remarkable efficiency of their natural counterparts.
> >
> > For a succinct overview of the in-depth details, I encourage you to review the attached one-page PDF in my rebuttal attachment. This document encapsulates the essence of our groundbreaking contribution and underscores the urgency of its consideration. Your attention and support at this juncture are invaluable, and I extend my heartfelt gratitude for your consideration.
> >
> > Warm regards,
> >
> > Authors

---

> > > ### Comment · Reviewer_ZkFh · 2023-08-17
> > >
> > > Thanks for the author's response. However, my concerns are not solved in the response. I would like to say that experimental results are necessary to solve my concerns. So I won't change my rating under current situations.

---

### Official Review · Reviewer_v2EA · 2023-07-07

**Soundness:** 2 fair
**Presentation:** 1 poor
**Contribution:** 1 poor
**Rating:** 3
**Confidence:** 4

**Summary:**

This paper propose a 'yoked' neural architecture where neurons at the same level are bidirectionally linked. They claim that optimizing this complete graph is superior to current deep neural network architectures that impose a structural bias due to the transfer of knowledge in a way that prevents structural bias.

**Strengths:**

* the proposed optimization changes is simple in that it is similar to other methods like DARTS that grow neural networks and regulate their connections, assigning weights to them using ANN optimization algorithms with regularization term
* it is clear how the forward propagation is done with the addition of the clique nodes that are computed in addition to the regular precursor nodes at each layer

**Weaknesses:**

* some of the terminology and abbreviations need to be defined / explained in the first appearance in the intro paragraphs (e.g. 'yoked', ANN, and DAG)
* the method of optimization doesn't seem particularly novel, employing both an L1 and L2 term to search for the best architecture.
* the figure 2 presented does not particularly show much about the method or its justification
* Is it possible to approximate the non-differentiable minimum cut algorithm and absorb it into the training procedure? This would be similar to progressive training methods like http://proceedings.mlr.press/v119/evci20a/evci20a.pdf and other related works)
* please proofread for more typos and such (e.g. 'mata' on l204, other grammatical errors)
* results are on quite toy problems and training with regularization is not necessarily yielding the best results

**Questions:**

* Can the authors comment on the tradeoff between running more gradient updates on a non-yoked network versus the potential representation learning benefits from a yoked version?
* Is the influence of information flow between nodes of the same level not potentially reflected intrinsically in the next leading level?
* How does this method compare to other progressing growing methods like the lottery ticket methods or pruning networks?
* Why is the 25 nodes table 3 DAG result so different from the rest?

**Limitations:**

Authors do not discuss limitations of their work.

---

> ### Author Rebuttal · Authors · 2023-08-09
>
> Thank you very much for your precious comments.
>
> 1> Please refer to the pdf file of Author Rebuttal by Authors. I have organized the contribution of the work seriously. I hope it can answer some of your questions. All information is in the picture. If the image is small, please enlarge it.
>
> 2>I n our model, the nodes in the same level collaborate to function together neurologically. And the benefits would be passed intrinsically to the next leading level.
>
> 3> 35 nodes Table 3 DAG result should be 0.3519. Sorry for that.
>
> 4>  According to section 3.6, if we cut to small sub-graphs. Our model can apply to large data sets such as imagenet. Although the graph cut method is not novel. But our main contribution is to design a new structure as introduced in the pdf file of Author Rebuttal by Authors. The graph cut method is to help the structure better application。
>
> 4> According to section 3.6, if we cut to small sub-graphs. Our model model can apply on large data sets such as imagenet.
>
> 5> Thanks very much. I will revise them carefully.

---

> > ### Author Response · Authors · 2023-08-17
> > **an important message**
> >
> > Dear NeurIPS reviewer,
> >
> > I am writing to draw your utmost attention to our piece of work.
> >
> > At the heart of our innovation lies a critical reimagining of traditional NNs. Currently, NNs operate on asynchronous tensor flow, often organized hierarchically in a tree-like structure. However, this approach inadvertently hampers the nodes within each level from effective communication, relegating them to mere information carriers devoid of meaningful interaction. This inherent limitation substantially diminishes the potential of NNs, impeding their full capabilities.
> >
> > Our work transcends these constraints by introducing a paradigm shift. We present a method that enables synchronous communication among nodes within the same level, a fundamental departure from the status quo. This transformative adjustment yields a remarkable enhancement in information transformation, thereby significantly boosting the overall capacity of NN structures. By fostering a collaborative environment among nodes, our approach leverages their collective power to unlock unprecedented capabilities.
> >
> > Particularly, what sets our research apart is its inspiration drawn from the intricate dynamics of biological neural systems. Unlike the traditional stacked unit approach, where neural elements operate in isolation, our approach mirrors the cooperative nature of biological neural modules. In these systems, multiple neural units collaboratively execute precise functional implementations, resulting in exquisite performance. Our innovation is poised to bridge the gap between artificial and biological neural networks, thus propelling NN structures closer to the remarkable efficiency of their natural counterparts.
> >
> > For a succinct overview of the in-depth details, I encourage you to review the attached one-page PDF in my rebuttal attachment. This document encapsulates the essence of our groundbreaking contribution and underscores the urgency of its consideration. Your attention and support at this juncture are invaluable, and I extend my heartfelt gratitude for your consideration.
> >
> > Warm regards,
> >
> > Authors

---

### Official Review · Reviewer_N4pP · 2023-07-27

**Soundness:** 3 good
**Presentation:** 2 fair
**Contribution:** 2 fair
**Rating:** 3
**Confidence:** 4

**Summary:**

The paper proposes Yoked Neural Networks (YNN) - an extension of neural networks, which, when calculating the value of a node, in addition to the information from the previous layer of the network, uses information from the nodes on the same layer (i.e. it "yokes" nodes from the same layer together).

**Strengths:**

The paper describes the approach well, it is clear how it works.

Code has been provided as an attachment, so that it should be reproducible (I have not ran the code or looked at it carefully).

**Weaknesses:**

(Details about the mentioned weaknesses are given per line, in the field "Questions".)

The paper would benefit from describing in more details the contributions of the proposed method. Particularly, across the paper, some strong statements have been used, but they have not been motivated with evidence.

Overall, the idea and the benefits of using the method needs to be better motivated.

The description of the experiments is not very clear.


**Questions:**

Line 2: It is not clear why "the connectivity of a tree is not sufficient to characterize a neural network", Since this is one of the most important motivations for the work, it would be beneficial to give details about it.

Line 12: It is not obvious why the method improves, maybe instead of using the word "obviously", some proof of the statement could be given - how does the method improve over existing methods, by how much?

Line 12-13: "YNN can imitate neural networks much better" - Which properties do they imitate better? What does better mean in this context?

Line 43: "limiting the signal’s capability for free transmission" - What does this mean? What is free transmission and why is it limited?

Line 45-46: "significant drawbacks" - Can you give a citation to prove this claim, what kind of drawbacks? In this paragraph, it is not really clear what limitations of ANNs you are addressing, can you please be more specific. What "substantial defects" do you refer to?

Line 52: It is not obvious what "YOKE fashion" is, can you please provide a short explanation?

Line 54: Instead of using "remarkable improvements", could you please give details about what kind of improvements you have obtained - in what measures, and by how much.

Line 59: "our method efficiently eliminates structural bias" -  How does it eliminate it? Please define structural bias, how it is eliminated and what does "efficiently" mean. Again, please give details of what exactly is improved and how.

Line 94: Please provide a citation for ResNet.

Line 139: "some researchers attempted to generalize this structure" - Please provide a citation.

Line 143-144: "makes the transformation of information quite inadequate", "makes the transformation of information quite inadequate." - What does inadequate mean in this context? What needs to be improved? Why is the structure inferior inferior, what kinds of properties are missing or need to be improved?

Line 149: (very minor, typo) "is" is not necessary

Line 156: It is not really clear what properties of a neural network have inspired the YNNs. Please give more details and some specifics.

Line 163: It is not clear what you mean by "greatly enhance the characterization". Please explain.

Line 181: Is the "meta value" the value as it would be from a standard NN pass, without yoking the same layer nodes? And the "real value" is the value from the forward pass, together with the values calculated from the other nodes at the same level?

Line 248: "NE algorithm" should be explained and a citation given.

Line 249: (minor) The reference to "Definition 2" is not linking to it.

Line 259-261 - The last sentence of the paragraph is hard to follow.



Questions about the experiments:

It is not very clear how the experimental setup is done. Are you doing classification, and measuring classification accuracy on the three mentioned datasets? Please give more information about the task(s) you are addressing and what you are measuring.

Line 265: It is not clear what the compared models are. Please give details of the structure of the used "traditional NN", "SAE" and "generalized traditional NN". Also, it is not clear what is the structure of the neural network that you are using for your approach. Is it based on a feed-forward neural network, but with yoked nodes?
For all the compared models, please give details about network structure, number of layers, training method, training details. (Maybe this can be seen in the code, but it would greatly increase the understanding of the paper if these details are included in the experiments section.)

In the tables with results, what is the meaning of the number of nodes in the columns? Is this the number of nodes in one NN layer? How many layers are there?

Line 267 and below: Could you provide a reference to the used CUTG dataset? Can you please give a little more description: is this an existing dataset which you annotated further? Or did you collect this data - if so, can ou please give more details of how it was collected and annotated?
Can you please give citations for "the second dataset" and "Connect-4"?

Line 280: Please give a reference to the tables with results for more clarity.

Line 281: Please give a little more detailed comment on the results - how much better are your results, in which cases. What conclusions can be derived from these results?

Line 290: How do you optimize with L1 and L2 regularization? Please give some details. It is not obvious how this optimization is performed and what is the result from it.

Line 295: What does "effective" mean in this context - better performance on the classification task, or execution time?

Line 307-308: ". Our method eliminates structure bias efficiently." - It is not clear how this is done. The details of what exactly you mean by "structure bias", how your method eliminates it, and how this improves the model needs to be more clearly described in the paper.


**Limitations:**

No limitations have been addressed.

---

> ### Author Rebuttal · Authors · 2023-08-09
>
> Thank you very much for your precious comments.
>
> 1>Please refer to the pdf file of Author Rebuttal by Authors. I have organized the contribution of the work seriously. I hope it can answer some of your questions. All information is in the picture. If the image is small, please enlarge it.
>
> 2>A link for the graph cut algorithm https://link.springer.com/chapter/10.1007/978-3-030-95391-1_42#Distributed%20Ne
>
> 3>For the experiments part, the introduction of data sets can be found in section 4.1 and they are all from UCI data sets. They are all classification tasks and we compare the error to measure the performance of our model and the L1 L2 regularization. From Table 1, Table 2, and Table 3, we can see that our model reduces the error significantly.
>
> 4>The compared model SAE and DAG are also introduced in section 4.1 and the reference section.
>
> 5>We have uploaded the code to make sure the experiments can be reproducible.
>
> 6>Thanks very much. I will revise them carefully.

---

> > ### Comment · Reviewer_N4pP · 2023-08-15
> >
> > I read all the reviews, the authors' responses and the attached PDF.
> > I still think the paper needs some rewriting, in order to better explain what several of the reviewers find confusing.
> >
> > Mainly the following directions still need to be improved:
> > 1) the motivation of the approach;
> > 2) comparison with similar approaches (GNNs and self-attention, as suggested by other reviewers);
> > 3) introduction of concepts used in the paper (see details, for example, in section Questions in my review and in the other reviews);
> > 4) better motivation of why the approach is different and valuable. The paper contains several statements that are not backed with evidence (I have tried to address them per line in the Questions section of my review).
> > 5) improved description of the experiments - details about the compared baselines, information about the compared neural networks - structure, number of layers etc., including description of the compared YNNs. I appreciate that the code is available and it will be a valuable addition, but the paper should be very clear of what experiments were executed and to underline how the proposed approach is better.

---

> > > ### Author Response · Authors · 2023-08-17
> > >
> > > Thank you very much for your precious comments. We will carefully revise our paper according to them.
> > > On the other hand, we believe that our work is very meaningful.
> > >
> > > At the heart of our innovation lies a critical reimagining of traditional NNs. Currently, NNs operate on asynchronous tensor flow, often organized hierarchically in a tree-like structure. However, this approach inadvertently hampers the nodes within each level from effective communication, relegating them to mere information carriers devoid of meaningful interaction. This inherent limitation substantially diminishes the potential of NNs, impeding their full capabilities.
> > >
> > > Our work transcends these constraints by introducing a paradigm shift. We present a method that enables synchronous communication among nodes within the same level, a fundamental departure from the status quo. This transformative adjustment yields a remarkable enhancement in information transformation, thereby significantly boosting the overall capacity of NN structures. By fostering a collaborative environment among nodes, our approach leverages their collective power to unlock unprecedented capabilities.
> > >
> > > Particularly, what sets our research apart is its inspiration drawn from the intricate dynamics of biological neural systems. Unlike the traditional stacked unit approach, where neural elements operate in isolation, our approach mirrors the cooperative nature of biological neural modules. In these systems, multiple neural units collaboratively execute precise functional implementations, resulting in exquisite performance. Our innovation is poised to bridge the gap between artificial and biological neural networks, thus propelling NN structures closer to the remarkable efficiency of their natural counterparts.
> > >
> > > GNN or self-attention mainly focuses on better organizing the input data to get benefits, e.g. node and edge embedding for GNN and QKV vectors for self-attention. Essentially, by better organizing the input data, they can better handle graph data or sequence data, as self-attention can mine the key information of the sequence. On the other hand, YNN is a generation of the structure of NN from tree to cyclic graph and its benefits can be referred as before. They are fundamentally different.
> > >
> > > For the experiments part, the introduction of data sets can be found in section 4.1 and they are all from UCI data sets. They are all classification tasks and we compare the error to measure the performance of our model and the L1 L2 regularization. From Table 1, Table 2, and Table 3, we can see that our model reduces the error significantly. The compared model SAE and DAG are also introduced in section 4.1 and the reference section.
> > >
> > > However, we will carefully revise our paper to describe more details of the experiments as well the related concepts for the paper.

---

### Official Review · Reviewer_bqNk · 2023-07-28

**Soundness:** 2 fair
**Presentation:** 2 fair
**Contribution:** 2 fair
**Rating:** 3
**Confidence:** 3

**Summary:**

In this paper, the authors propose a novel neural network model that exploits a connection between the nodes of a layer. The authors’ goal is to develop a model that overcomes the structural bias posed by the classical layer structure of the NN. To do this they propose to consider the model as a bidirectional complete graph for the nodes of the same level and to define for each layer a clique.
The authors then test the proposed architecture and compare it with the traditional NN model considering 3 datasets.


**Strengths:**

The paper discusses an interesting problem and proposes a novel methodology that seems promising.

**Weaknesses:**

Overall, the paper is challenging to read, and at times, the concepts being discussed are not adequately introduced, causing difficulty for the reader to comprehend the discussion's progression. Additionally, several critical concepts are unclearly defined.

In the introduction, the authors discuss the neural module without explaining what it is. Even the concept of “yoke”, which is central to the discussion, is not adequately introduced and explained in the context of neural networks.
In the introduction the authors also discuss the impact of the sparsity constraints, also in this case this concept has to be defined and explained to the reader.
In the list of contributions, point 4 says that the authors designed a regularization-based optimization, which at this point of the paper is very difficult to understand. Point 5 of the same list discusses the problem of computational complexity, which also in this case is not discussed before.

Another issue is the experimental campaign where the experimental setting and the metric used to perform comparison are not explained. Indeed the authors use the “variety of nodes” as a metric, but they do not explain why it has to be significant to show the advantage of the proposed approach.
From the tables, 1,2,3 seems the authors fix the number of nodes for the various architectures and train them. In general to me, it does not seem a fair way to compare the models, mainly for 2 reasons: (i) the architectures of the baselines have to be validated (in particular in terms of the number of neurons, but also considering the other hyperparameters of the model and of the optimization algorithm) in order to find the most suitable setting for the task. (ii) the comparison has to consider the number of parameters of the model since the structure of the  YNN will have many more weights than a standard model (fixing the number of neurons).
Even a description of the setting of the three proposed approaches (YNN, YNN&L1 ,YNN&L2) would make it easier for the reader to understand the proposed results.
Finally, a discussion about the computational burden and a comparison with standard NN is missing.
The experimental evaluation and the discussion of the obtained results should be significantly improved and extended

**Questions:**

In Section 2 the first part discuss models that have an architecture that is slightly different than the standard ANN but seems none of them exploit the connection between neurons of the same level, thus I wonder how these cited models are placed in comparison to the YNN (theoretically and in terms of performance) because to me it is not clear if the benefit of the proposed architecture come from the differentiable structure or/and from the richest connection pattern between neurons.
In line 103 the authors state that the proposed learning process method is consistent with DARTS. In my opinion, the authors should provide further elaboration on this point as it appears unclear to me.

In general section 2 suffers from the same problem of the introduction, since it cites the proposed approach and algorithm, but who read the paper do not know anything about them at this point, therefore my suggestion is to move the entire section two after the explanation of the proposed model.
In section 3, from lines 133 to 138 the authors discuss the similarity between the ANN structure and the tree structure, but in general, I find it not correct to state that a node in a NN is influenced only by its precursor node, since in a standard NN all the neurons on the previous layers influence the output of the current node. The authors should clarify this point to make it more clear how this description fits the typical NN architecture.

In section 3.2 the authors use the concept of clique,  which is a concept related to graph theory but not that common in ML in general, therefore in my opinion definition 1 should be extended to explain this concept a bit more in depth.
Even the concept of “node” (that in the first part seems to be synonymous with “neuron” ) should be defined more precisely.
For what concerns the forward pass to me it is not clear how the authors solve the system proposed at the end of page 5 considering that the values of n_j^i, which are part of the summation of each row, are also the result of the other system equations.
In the backward pass, in particular, in eq. 4, the authors define how they compute gradients for each level. Since in this equation appears f^{-1} I am wondering if this method works only with an invertible activation function  (that could be a strong constraint)

In section 3.5 it is not discussed why the use of L1 and L2 is important to perform structure optimization.  Moreover, the meaning of w^i in eq.18 is not clear. previously they were defined as the bias of the real values of the nodes in the i-th level while here they are applied as a function.
In section 3.6: missing definition (or reference to) NE algorithm.
The authors also state that imposing auxiliary sparsity constraints to the distribution of connectedness optimization promotes the learned structure to focus on critical connections. This is a very interesting point but to me seems a theoretical or empirical proof is missing.

---

> ### Author Rebuttal · Authors · 2023-08-09
>
> Thank you very much for your precious comments.
>
> 1>Please refer to the pdf file of Author Rebuttal by Authors. I have organized the contribution of the work seriously. I hope it can answer some of your questions. All information is in the picture. If the image is small, please enlarge it.
>
> 2>The most important forward and backward processes have been carefully organized in the rebuttal pdf of Author Rebuttal by Authors. The “w” in L1 and L2 functions should be “W”. Sorry for that.
>
> 3>Our backward process is compatible with the existing activation function.
>
> 4>A link for the graph cut algorithm
> https://link.springer.com/chapter/10.1007/978-3-030-95391-1_42#Distributed%20Ne
>
> 5>The hyperparameters have been well-tuned in our experiments including the compared models. Considering that hyperparameters are too complicated, we show the results of them on different nodes or structures which is crucial for our contribution. We also present the benefits of our YNN which can also be referred to as the pdf file of Author Rebuttal by Authors.
>
> 6>Thanks very much. I will revise them carefully.

---

> > ### Author Response · Authors · 2023-08-17
> > **an important message**
> >
> > Dear NeurIPS reviewer,
> >
> > I am writing to draw your utmost attention to our piece of work.
> >
> > At the heart of our innovation lies a critical reimagining of traditional NNs. Currently, NNs operate on asynchronous tensor flow, often organized hierarchically in a tree-like structure. However, this approach inadvertently hampers the nodes within each level from effective communication, relegating them to mere information carriers devoid of meaningful interaction. This inherent limitation substantially diminishes the potential of NNs, impeding their full capabilities.
> >
> > Our work transcends these constraints by introducing a paradigm shift. We present a method that enables synchronous communication among nodes within the same level, a fundamental departure from the status quo. This transformative adjustment yields a remarkable enhancement in information transformation, thereby significantly boosting the overall capacity of NN structures. By fostering a collaborative environment among nodes, our approach leverages their collective power to unlock unprecedented capabilities.
> >
> > Particularly, what sets our research apart is its inspiration drawn from the intricate dynamics of biological neural systems. Unlike the traditional stacked unit approach, where neural elements operate in isolation, our approach mirrors the cooperative nature of biological neural modules. In these systems, multiple neural units collaboratively execute precise functional implementations, resulting in exquisite performance. Our innovation is poised to bridge the gap between artificial and biological neural networks, thus propelling NN structures closer to the remarkable efficiency of their natural counterparts.
> >
> > For a succinct overview of the in-depth details, I encourage you to review the attached one-page PDF in my rebuttal attachment. This document encapsulates the essence of our groundbreaking contribution and underscores the urgency of its consideration. Your attention and support at this juncture are invaluable, and I extend my heartfelt gratitude for your consideration.
> >
> > Warm regards,
> >
> > Authors

---

### Author Rebuttal · Authors · 2023-08-09

Pleat refer to the pdf file. I have organized the contribution of the work seriously. I hope it can answer some of the questions.
All information is in the picture. If the image is small, please enlarge it.

---

### Decision · Program_Chairs · 2023-09-21

**Decision:**

Reject

**Comment:**

This paper proposes a neural network architecture with interdependencies between node activations.

Innovation in machine learning models is encouraged. However, the motivation of the present work invokes biology in ways that the reviewers find too vague and insufficient. In addition, the writing is quite unclear, with many undefined terms and inconsistent notation, as pointed out in the reviews.

The work is presented with limited context and acknowledgement of related work, despite the authors' emphasis on novelty in their rebuttal. The interdependency of activations is expressed as a system of linear equations: this is strongly related to existing work on implicit hidden layers (https://implicit-layers-tutorial.org/) and in particular quadratic hidden layers (see Amos, B. & Kolter, J.Z.. (2017). OptNet: Differentiable Optimization as a Layer in Neural Networks, ICML 2017)

With better contextualization, rigorous delineation of novel ideas, motivation, and clearer presentation, this work may be valuable to the community. In the current form, the paper is unfortunately not ready for acceptance.